**Extensional reactivation of the Penninic Frontal Thrust 3 Ma ago as**
**evidenced by U-Pb dating on calcite in fault zone cataclasite.**
Antonin Bilau[a,b], Yann Rolland[a,b], Stéphane Schwartz[b], Nicolas Godeau[c], Abel Guihou[c], Pierre
Deschamps[c], Benjamin Brigaud[d], Aurélie Noret[d], Thierry Dumont[b], and Cécile Gautheron[d].
[a]EDYTEM, Université Savoie Mont Blanc, CNRS, UMR 5204, Le Bourget du Lac, France.
[b]ISTerre, Université Grenoble Alpes, Univ. Savoie Mont Blanc, CNRS, IRD, IFSTTAR, 38000
Grenoble, France.
[c]Aix-Marseille Université, CNRS, IRD, INRAE, Collège de France, CEREGE, Aix en Provence,
France.
[d]GEOPS, CNRS, Université Paris-Saclay, 91405 Orsay, France.
**Correspondence:** Antonin Bilau (antonin.bilau@univ-smb.fr) and Yann Rolland
(Yann.Rolland@univ-smb.fr).
**Abstract**
In the Western Alps, the Penninic Frontal Thrust (PFT) is the main crustal-scale tectonic structure of the
belt. This thrust transported the high-pressure metamorphosed internal units over the un-metamorphosed
European margin during the Oligocene (34-29 Ma). Following the propagation of the compression
toward the European foreland, the PFT was later reactivated as an extensional detachment associated
with the development of the High-Durance extensional fault system (HDFS). This inversion of tectonic
displacement along a major tectonic structure has been widely emphasized as an example of extensional
collapse of a thickened collisional orogen. However, the inception age of the extensional inversion
remains unconstrained. Here, for the first time, we provide chronological constraints on the extensional
motion of an exhumed zone of the PFT by applying U-Pb dating on secondary calcites from a fault zone
cataclasite. The calcite cement/veins of the cataclasite, formed after the main fault slip event, at 3.6±0.4-
3.4±0.6 Ma. Cross-cutting calcite veins featuring the last fault activity are dated at 2.6±0.3-2.3±0.3 Ma.
$\delta^{13}C$ and $\delta^{18}O$ fluid signatures derived from these secondary calcites suggest fluid percolation from deep-
seated reservoir at the scale of the Western Alps. Our data evidence that the PFT extensional reactivation
initiated at least ~3.5 Ma ago with a reactivation phase at ~2.5 Ma. This reactivation may result from
the westward propagation of the compressional deformation toward the External Alps, combined to the
exhumation of External Crystalline Massifs. In this context, the exhumation of the dated normal faults
is linked to the eastward translation of the HDFS seismogenic zone in agreement with the present day
seismic activity.

## 1. Introduction

Dating of major tectonic inversions in orogens is generally achieved by indirect and relative dating, but rarely by the direct dating of fault-related minerals using absolute geochronometers. For instance, tectonic cycles are defined worldwide by the sediment unconformities or by exhumation ages through thermochronological investigation. However, the recent progress in U-Pb dating of carbonate using high-resolution Laser Ablation analyses (Roberts et al., 2020) allows us to directly date minerals formed during fault activity and thus to establish the age of tectonic phases by absolute radiometric dates (Ring and Gerdes, 2016; Goodfellow et al., 2017; Beaudoin et al., 2018;). This method is especially well suited to disentangle the successive tectonic motions along a given tectonic structure. U-Pb dating can be coupled to stable isotopic analysis to infer the nature of fluids through time, which may give insights of the scale of fluid circulations and thus the scale of the active tectonic structure and changes in the stress regime (e.g., Beaudoin et al., 2015; Rossi and Rolland, 2014). In the Western Alps, the Penninic Frontal Thrust or PFT represents a major thrust structure at lithospheric scale (e.g., Tardy et al., 1990; Mugnier et al., 1993; Zhao et al., 2015) that accommodated the main collisional phase during the Paleogene-Neogene (e.g., Ceriani et al., 2001; 2004). Later on, this thrust was reactivated as a normal fault, and the extensional deformation is still ongoing (Sue and Tricart, 1999; Tricart et al., 2006; Sue et al., 2007). This transition from compression to extension in a collisional chain has been diversely interpreted to reflect slab breakoff, crustal overcompensation or post-glacial and erosion-induced isostatic rebound (e.g., Champagnac et al., 2007; Sternai et al., 2019). However, until now, no direct dating of the tectonic shift from compression to extension on the PFT has been obtained, which leads to many possible geodynamic scenari. At the present day, a large range of ages for this transition has been hypothesized from ~12 to 5 Ma (Tricart et al., 2006), to only few ten's ka (Larroque et al., 2009) which shows the lack of direct dating of brittle deformation (Bertrand and Sue, 2017). In this study, we applied the Laser Ablation U-Pb dating method on secondary calcites from a cataclasite fault zone that testify of the extensional deformation of an exhumed paleo-normal fault during the PFT inversion.

The purpose of this study is (1) to provide absolute chronological constraints on the structural inversion of the PFT, and (2) give insights into the scale and nature of fluid circulations along this major fault using stable isotope analysis of carbon and oxygen.

## 2. Geological setting

The western Alpine collisional belt results from the convergence and collision of the European and Apulian plates, which culminated with top-to-the west displacement on the PFT acting as the major Alpine tectonic structure in the Late Eocene to Oligocene times (e.g., Dumont et al., 2012; Bellahsen et al., 2014). This lithospheric-scale structure accommodated westward thrusting of highly metamorphosed "Internal zone" units over slightly metamorphosed "External zone" units (Fig. 1, Schmid and Kissling

2000; Lardeaux et al., 2006; Simon-Labric et al., 2009; Malusà et al., 2017). The External zone is composed of the European non-metamorphosed Mesozoic and Paleozoic sedimentary cover and its Paleozoic basement corresponding to the External Crystalline Massifs.

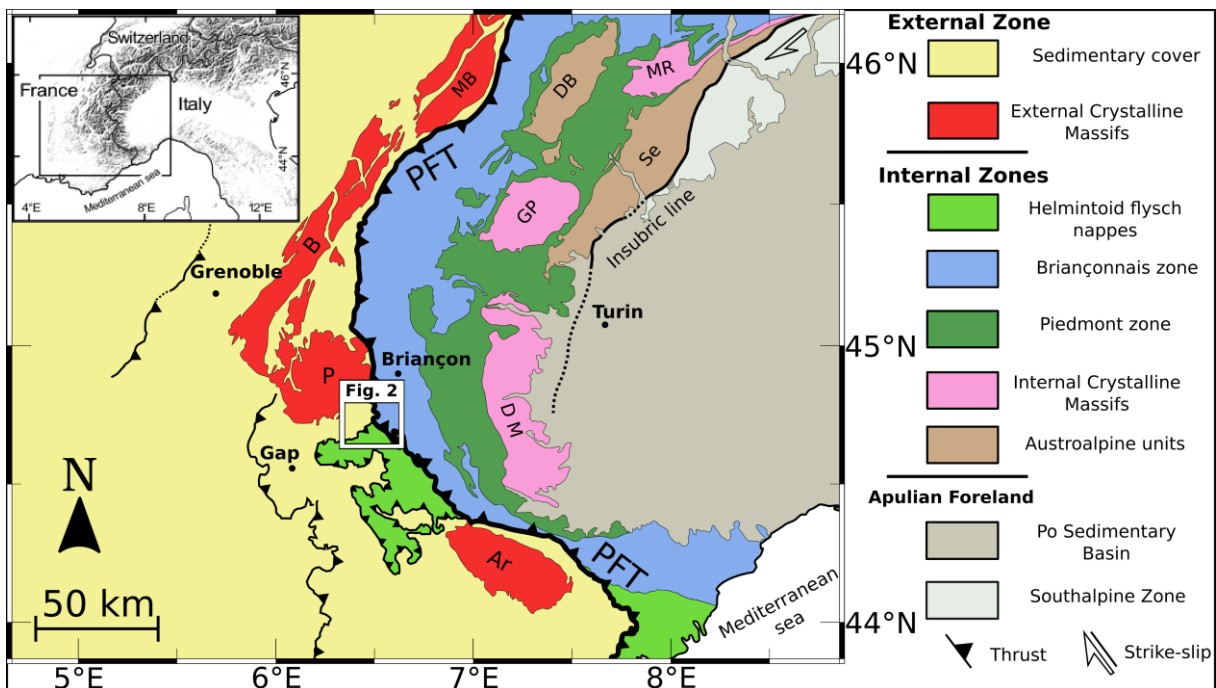

**Fig. 1.** *Geological map of Western Alps showing the location of the study area. External Crystalline Massifs: Ar, Argentera; B, Belledonne; MB, Mont Blanc; P, Pelvoux. Internal Crystalline Massifs: DM, Dora-Maira; GP, Grand Paradis; MR, Mont Rose. PFT: Penninic Frontal Thrust. Insert modified from Schwartz et al. (2017).*

The Internal zone corresponds to a high-pressure metamorphic wedge formed by the stacking of the paleo-distal European margin of the Briançonnais zone, comprising the Internal Crystalline Massifs and their sedimentary cover, with the oceanic-derived units of the Piedmont zone. These units were incorporated and juxtaposed in the subduction accretionary prism since the Early Late Cretaceous until the Late Eocene (e.g., Agard et al., 2002; Schwartz et al., 2007). The timing of subduction and collision is well constrained by numerous dates on metamorphic minerals (e.g., Duchêne et al., 1997; Rubatto and Hermann, 2001; Lanari et al., 2012, 2014). Eclogite facies recrystallization records subduction of the distal European margin at $32.8 \pm 1.2$ Ma in the Dora Maira massif, which was later transported as a tectonic nappe during the collision (Duchêne et al., 1997). PFT activation and underthrusting of External Crystalline Massifs are indicators of the transition from subduction to continental collision in the Internal zones, between 44 and 36 Ma (e.g., Beltrando et al., 2009). This transition is marked by shear zone development at greenschist facies conditions and recrystallization during burial of the Alpine External zone in the PFT footwall compartment (Rossi et al., 2005; Sanchez et al., 2011; Bellahsen et al., 2014). The early ductile PFT activity is dated at 34-29 Ma by $^{40}Ar/^{39}Ar$ dating of syn-kinematic phengite from shear zones in the Pelvoux and Mont Blanc External Crystalline Massifs (Seward and Mancktelow,

1994; Rolland et al., 2008; Simon-Labric et al., 2009; Bellanger et al., 2014; Bertrand and Sue, 2017)
and by U-Pb on allanite (Cenki-Tok et al., 2014). The age of the PFT hanging wall tectonic motion and
joint erosion is highlighted by the exhumation of the Briançonnais units constrained by apatite fission
tracks (AFT) at 26-24 Ma (Tricart et al., 1984, 2001, 2007; Ceriani and Schmid, 2004). However, the
PFT reactivation as a normal fault remains unconstrained. The onset of PFT extensional activity has
been proposed to the Late Miocene (~12 to 5 Ma), based on indirect AFT ages in the Pelvoux External
Crystalline Massif (Tricart et al., 2001, 2007), that record a cooling episode related to relief creation and
erosion. The current seismicity (e.g., Rothé, 1941; Sue et al., 1999, 2007) and observed GPS motions
(Walpersdorf et al., 2018; Mathey et al., 2020), all along the so-called High-Durance Fault System
(HDFS) highlight the fact that extensional and minor strike-slip deformations along the PFT are still
ongoing. This seismicity mostly occurs at shallow depths, less than 10 km, and mainly at 3 to 8 km,
where the HDFS is structurally connected to the PFT (Sue and Tricart 2003, Thouvenot and Fréchet,
2006; Sue et al., 2007).

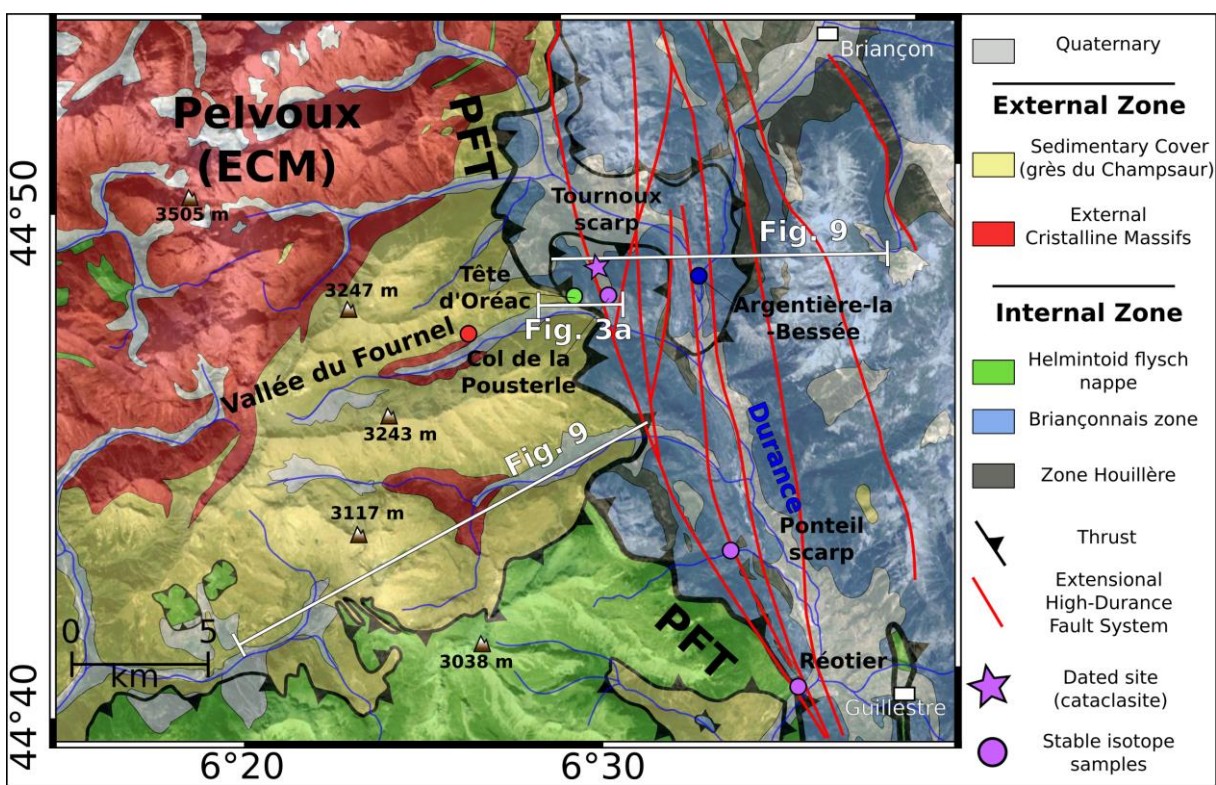

*Fig. 2. Study area of the Penninic Frontal Thrust, east of the Pelvoux External Crystalline Massif (ECM). High-*
*Durance Fault System is represented in red from Tricart et al. (2001) and Sue et al. (2007). Location of sampled*
*sites is indicated. The location of the extensional fault dated by U-Pb on calcite (samples FP18-2 and FP18-3) is*
*marked by a star. Colour of site circle refers to the host rock age: red, Eocene sandstone flysch (grès du*
*Champsaur); green, Cretaceous carbonates; blue, Jurassic carbonates; purple, Triassic carbonates. Sample*
*descriptions are shown on Suppl. Mat. 1. © Google Earth for background relief map.*

The study area is focused on a portion of the PFT located in the southeast of the Pelvoux External Crystalline Massif in the Western Alps (France) (Figs. 1-2). Here, the PFT rests on Late Eocene (Priabonian) autochtonous nummulitic flysch so-called the "Champsaur sandstone" (Fig. 2), which lies unconformably on the Pelvoux crystalline basement. In the southern part, the PFT lies on the Cretaceous Helmintoid flysch nappes, Fig. 2. These two flysch units are intensely deformed by top-to-the-west PFT compressional deformation. The PFT hanging wall corresponds to the Briançonnais zone composed of Mesozoic and Paleozoic sedimentary units, which underwent high pressure metamorphism (Lanari et al., 2012; 2014). The Briançonnais zone is composed of the Briançonnais Zone Houillère, which consists of Carboniferous sediments overlying a crystalline basement, stratigraphically overlain by Middle Triassic to Cretaceous sediments (limestones and calcschists). The PFT structure is well shown in the Tête d'Oréac section of the Fournel Valley transect (Fig. 3, Sue and Tricart, 1999). Here, normal faults cross-cut the Briançonnais series and branch down on the PFT, which was reactivated as a detachment (Tricart et al., 2001).

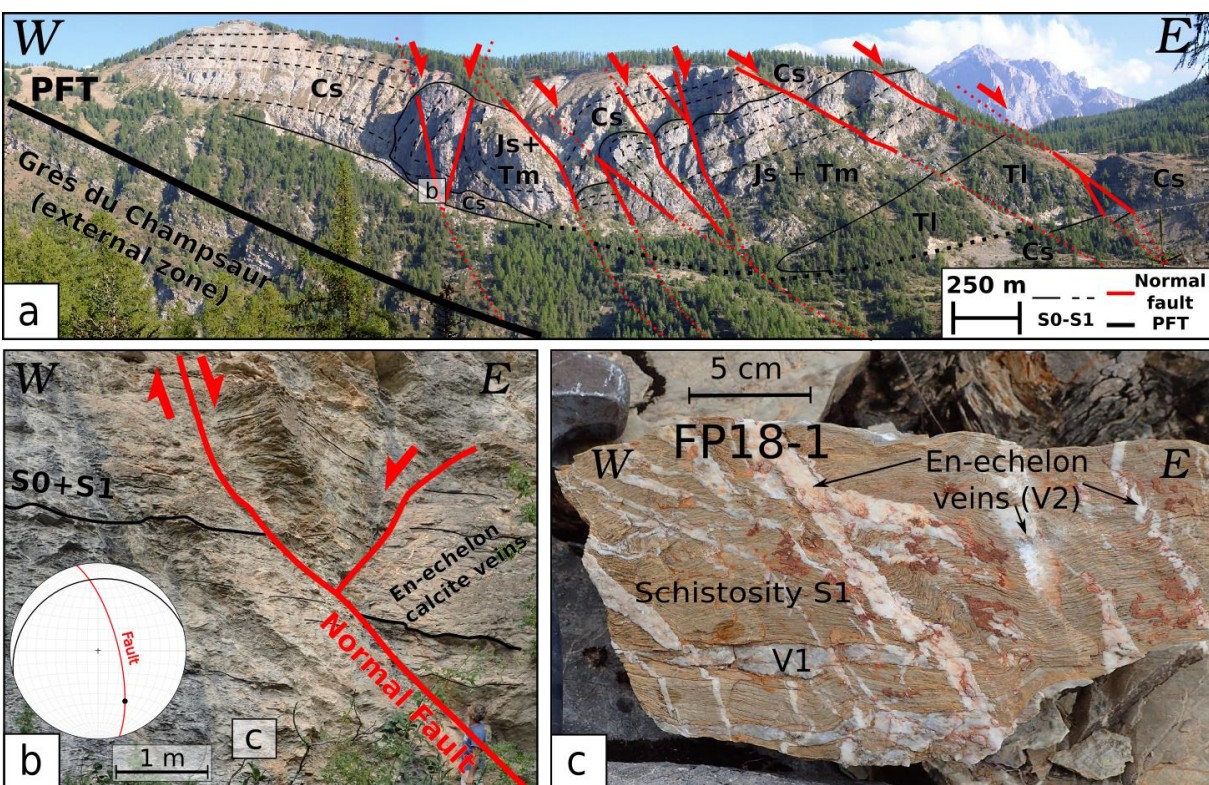

***Fig. 3. a:*** *General view and geological interpretation of the Fournel Valley southern slope with the studied site of the Tête d'Oréac.* ***b:*** *Outcrop interpretation of the Tête d'Oréac with extensional features in late Cretaceous calcschists in agreement with the High-Durance Fault System and Wulff stereogram, lower hemisphere.* ***c:*** *Calcschist oriented sample FP18-1 evidencing multiple calcite vein generations. V1 is related to the main compressional phase related to the Tête d'Oréac anticline formation and V2 are related to extensional reactivation of the PFT during onset of the High-Durance Fault System. Cs: Late Cretaceous calcschists; Js+Tm: Middle Triassic to Late Jurassic dolomitic to siliceous limestones; Tl: Lower Triassic sandstones.*

The normal faults are tilted by a passive rotation of about 30 degrees towards the west during their
exhumation in relation with the activity of the High-Durance Fault System (Sue et al., 2007).
**3. Sampling strategy and analytical methods**
*3.1. Sampling strategy*
We collected key samples of each brittle-ductile deformation phase, both in the PFT footwall and
hanging wall (Suppl. Mat. 1), to provide a petrographic and stable isotopic dataset which will allow
discussing the nature of fluids throughout the PFT activity associated to the late compressional and
extensional history. Field analysis is supported by petrographic observations on 28 samples, including
8 host rocks, 6 from compressional structures and 14 from extensional structures. Based on this dataset,
we selected three fault breccia samples to date the PFT extensional reactivation.
*3.2. Cathodoluminescence*
Cathodoluminescence (CL) analysis provides shades that are mainly representative of oxidation state of
trace element and their contents, i.e. $Mn^{2+}$ and $Fe^{2+}$ (Barnaby and Rimstidt, 1989). These differences in
calcite chemical composition are an indicator of different mineral precipitations related to slight
variations in fluid composition (Goodfellow et al., 2017). CL can also highlight crystal growth patterns
or grain boundary interactions (Beaudoin et al., 2015). Using cross-cutting criteria as well as CL, a
relative chronology of the calcite generations and related microstructures has been made. Analyses were
performed with a spot camera mounted-Cathodyne device (cold cathode) with the following parameters:
vacuum ~50mTorr; voltage 16-18 kv; electron beam ~200 µA. Used description terminology is based
on Bons et al. (2012).
*3.3. O and C stable isotope analysis*
Stable isotope measurements were achieved on the different generations of microstructures identified
by thin section observations and CL images, at Geosciences Paris Sud (GEOPS) laboratory of the Paris-
Saclay University, France. Results are presented in Table 1. The protocol is described in detail by
Andrieu et al. (2015). Several milligrams (~$1mm^3$) of sample for each calcite generation were collected
using a Dremel 4000 with a 3.2 mm head. Samples were then dissolved with pure orthophosphoric acid
($H_3PO_4$): Sample tubes provided with two compartments (one for the sample and one for the acid) were
sealed under a pressure of $1.5x10^{-2}$ mbar. They were immersed in a water bath at 25°C before the acid
was poured on the sample and let to react for 24 h. Complete reaction is necessary to avoid any artificial
isotopic fractionation. The produced $CO_2$ is collected using an extraction line and a liquid nitrogen trap
is used to ensure that only $CO_2$ is collected. Pure $CO_2$ is analyzed on a VG Sira 10 dual inlet IRMS
(Isotope Ratio Mass Spectrometer). Data validity is supported by concurrent analysis of the international
standard IAEA CO-1. $\delta^{13}$C and $\delta^{18}$O are expressed in ‰ relative to V-PDB (Vienna Pee Dee Belemnite)
by assigning a $\delta^{13}$C value of +1.95‰ and a $\delta^{18}$O value of −2.20‰ to NBS19, (1).
$$\delta^{13}C = \left[ \frac{(^{13}C/^{12}C)_{Sample}}{(^{13}C/^{12}C)_{Reference}} - 1 \right] \times 1000 \ (1).$$
For oxygen isotope measurements, switch from PDB values to SMOW (Standard Mean Oceanic Water)
were made using the Kim et al. (2015) equation, (2).
$$\delta^{18}O_{SMOW} = 1.03086 \times \delta^{18}O_{PDB} + 30.86 \ (2).$$
The ratio of carbon and oxygen isotopes is related to the parental fluid of calcite and can be used as a
fluid tracer. Reproducibility was checked by replicate analysis of in-house standards and was ±0.2‰ for
oxygen isotopes and ±0.1‰ for carbon isotopes.

*3.4. U-Pb dating of calcite*
In-situ uranium and lead isotope analyses of carbonates were carried out at CEREGE (Centre Européen
de Recherche et d'Enseignement des Géosciences de l'Environnement), Aix-en-Provence, France.
Results are presented in Suppl. Mat. 2. Data were acquired on 150 µm thick thin sections. Laser ablation
analysis was performed with an ESI excimer Laser Ablation system with a 6 inches two volume cell
(ESI), coupled to an Element XR SF-ICP-MS (Sector Field Inductively Coupled Mass Spectrometer,
Thermo-Scientific). Analyses were done at 10 Hz and 1.1-1.15 J.cm$^{-2}$. Samples were first screened to
check signal intensities and maximise the spread of $^{238}$U/$^{206}$Pb ratios (e.g. map of Suppl. Mat. 3) to obtain
the highest U-Pb variability. A typical analysis consists of 3 seconds of pre-ablation to clean the sample
surface, followed by 20 seconds of gas blank and ~20 seconds of measurement on a static circle spot of
150 µm diameter (approximately 8-9 acquisition cycles per second). These parameters lead to
approximately ~20-25 µm depth hole (~1 µm/s) on a carbonate material. Ablated particles are carried
out of the cell with a He gas flux of 1300 ml/min and then mixed with Ar sample gas (typically 0.8-0.9
l/min). Unknown samples were corrected by standard bracketing with synthetic NIST-614 glass for
instrumental drift and lead isotope composition (Woodhead et al. 2001) and a natural calcite spar WC-
1 of 254.4 ± 6.4 Ma (Roberts et al., 2020) for inter-elemental fractionation effect, every 20
measurements. No downhole correction was applied since no natural calcite standard with homogeneous
U-Pb ratio allows such correction. However, the large aspect ratio used in this set up is supposed to limit
this effect. Unknown sample were first processed with the Iolite software (Paton et al., 2011) for baseline
correction. Raw ratios were then reduced for instrumental drift, lead isotope composition and inter-
elemental fractionation using an in-house excel spreadsheet macro designed for carbonate samples. Ages
are obtained using IsoplotR software and plotted in a Tera-Wasserburg diagram using model (1) age
(Vermeesch, 2018). An additional error propagation of 2.51% in quadratic addition on the final age, tied
to the WC-1 standard, is expressed in brackets in the Tera-Wasserburg plot.

## 4. Results

*4.1. Deformation phases and miscrostructures*

4.1.1. Brittle-ductile deformation features

During the westward thrust motion of the PFT, the Tête d'Oréac cross-section passes through the PFT (Fig. 3) and preserves a succession of units that were stacked on each other. The main schistosity (S1) is parallel to the initial bedding (S0) in Cretaceous calcschists. S0-S1 is sub-horizontal and penetrative throughout the studied area. At the outcrop scale, S1 is clearly visible and shows dissolution surface with the development of stylolithic joints (Fig. 4).

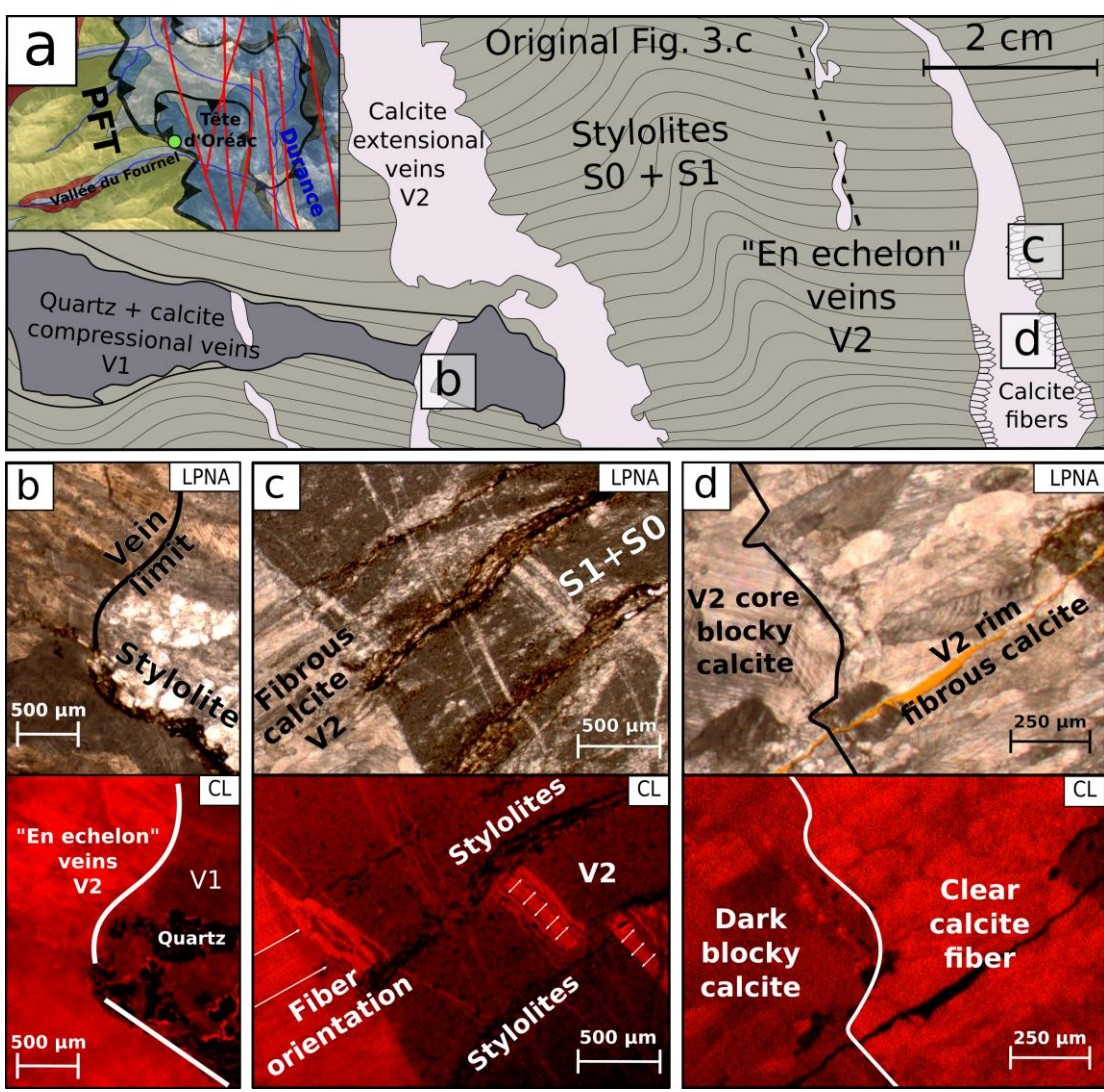

**Fig. 4. a:** *General sketch of sample FP18-1 evidencing cross-cutting relationships for two main vein generations fig. 3.c.* **b-d:** *Microscope and cathodoluminescence pictures showing the different vein calcite generations.*

Quartz anisotropy is observable in LPA which indicates an important deformation syn-post V1. This suggests a strong transposition of structures during PFT compressional motion or the veins opened initially in an orientation parallel to S1 either way a ductile deformation is recorded. These early

shortening features are cross-cut by numerous steeply dipping eastward normal faults linked to the
extensional reactivation of PFT. Early stages of extension are featured by centimetre scale "en-echelon"
veins (V2) indicative of an early brittle-ductile extensional deformation followed by dissolution on the
horizontal composite (S0-S1) cleavage. Larger V2 veins, expressed at centimetre scale, cross-cut the
cleavage and show elongated calcite fibres of ~1000 μm at the vein walls (Fig. 4). Similar shades for
early V2 and fibrous V2 are observed in CL. At vein cores, the fibrous calcite is then replaced by a
blocky calcite that is less luminescent in CL.

4.1.2. Brittle deformation features
The internal structure of one major extensional fault is investigated in the Tournoux scarp (Fig. 5). The
fault zone is highlighted by a metre-scale cataclasite fault gouge with variable amounts of deformations.
The top-to-the East (N90°E) normal sense of shear is represented by sigmoids and down-dip slickenside.
At thin-section scale, for sample FP18-2, the cataclasite is composed of centimetre-scale host rock clasts
with very small (<20 μm) limestone grains. Two types of calcite fillings have been identified. The first
one contains organic matter has a « dusty appearance » with bright shades in CL (Fig. 5C). The second
one shows large and clear crystals that grew in the cracks and porosity, showing sector zoning patterns
highlighted in CL and Laser Ablation-Inductively Coupled Plasma-Mass Spectrometry (LA-ICP-MS)
maps (Fig. 5D; Suppl. Mat. 3). ~700 μm large hexagonal, clear and organic matter free, calcite crystals
have been selected for U-Pb dating.
These calcite crystals represent the latest pervasive fluid circulation episode through the porosity and
provide a minimum age for the cataclasite. In sample FP18-3, the matrix is cross-cut by calcite veins
with variable diameters (300-1300 μm) and is free of any further deformation. On the basis of their
homogeneity and their youngest relative age relationships, these late calcites have also been targeted for
U-Pb calcite dating (see section 4.3). Samples FP19-12A-B (described in supplementary data) were
collected in a west-dipping conjugate normal fault and exhibits similar deformation features.

***Fig. 5. a-b:*** *Outcrop interpretation of the Tournoux scarp showing various degrees of cataclasis in Triassic*
*dolomitic limestone with Wulff stereogram lower hemisphere. Squares are sampled area, sample FP18-2 is a*
*highly cataclased sample, while sample FP18-3 is less intensely cataclased and is cross-cut by millimeter-scale*
*calcite veins.* ***c, d, e:*** *Microscope and cathodoluminescence pictures showing several calcite filling generations.*
*« clear calcite » shows zonings and seems to crystallize into a primary porosity left within the cataclasite. The*
*clear calcite and veins from the cataclasite are dated using the U-Pb dating on calcite method.*

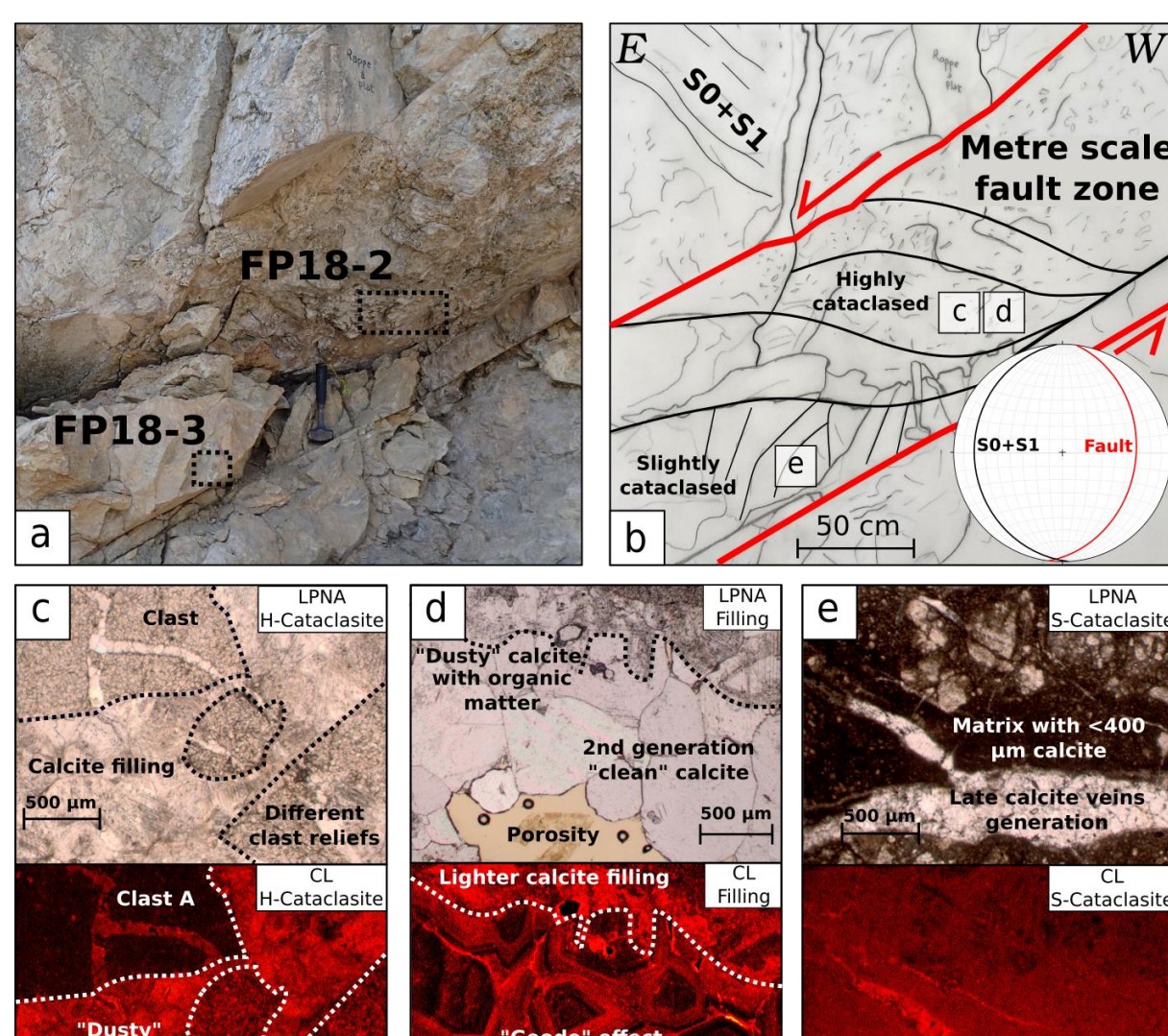


*4.2. $\delta^{13}C$ and $\delta^{18}O$ stable isotope results*

Stable isotopes analyses were performed in calcites from various host rocks samples belonging to the
different units highlighted in the studied PFT section (Fig. 3) and are supposed to be representative of
the different (compressional and extensional) key tectonic phases (Fig. 6).
For host rock analysis, upper Cretaceous planktonic calcschists from the Tête d'Oréac show the lowest
$\delta^{18}O$ host rock value of 16.8-17.1 ‰ and of $\delta^{13}C$ of 2.1-2.2 ‰. Triassic carbonates show a range between
23.7-26.5 ‰ for $\delta^{18}O$ and between 1.9-2.3 ‰ for $\delta^{13}C$ (with a higher value of 3.4 ‰ for the Ponteil
scarp). Upper Jurassic calcshists gave $\delta^{18}O$ ratio of 28.5 ‰ and $\delta^{13}C$ of 1.3 ‰. The western Late Eocene
Flysch (Champsaur sandstone) gave lowest $\delta^{13}C$ ratio of -0.3 ‰ and a $\delta^{18}O$ ratio of 21.9 ‰. Analysed
brittle-ductile veins either related to the compressional or to the onset of the extensional tectonic phases
stand very close to their host rocks, near to the meteoric water field defined by Nardini et al. (2019)
(Fig. 6). However, the V2 veins associated to the brittle normal fault development, clearly show lower
$\delta^{18}O$ values (<15‰) compared to their host rocks, with a trend towards lower $\delta^{13}C$ values. These isotope
signatures are similar to those measured in calcite from veins of the Mont Blanc External Crystalline
Massifs (Rossi and Rolland, 2014).

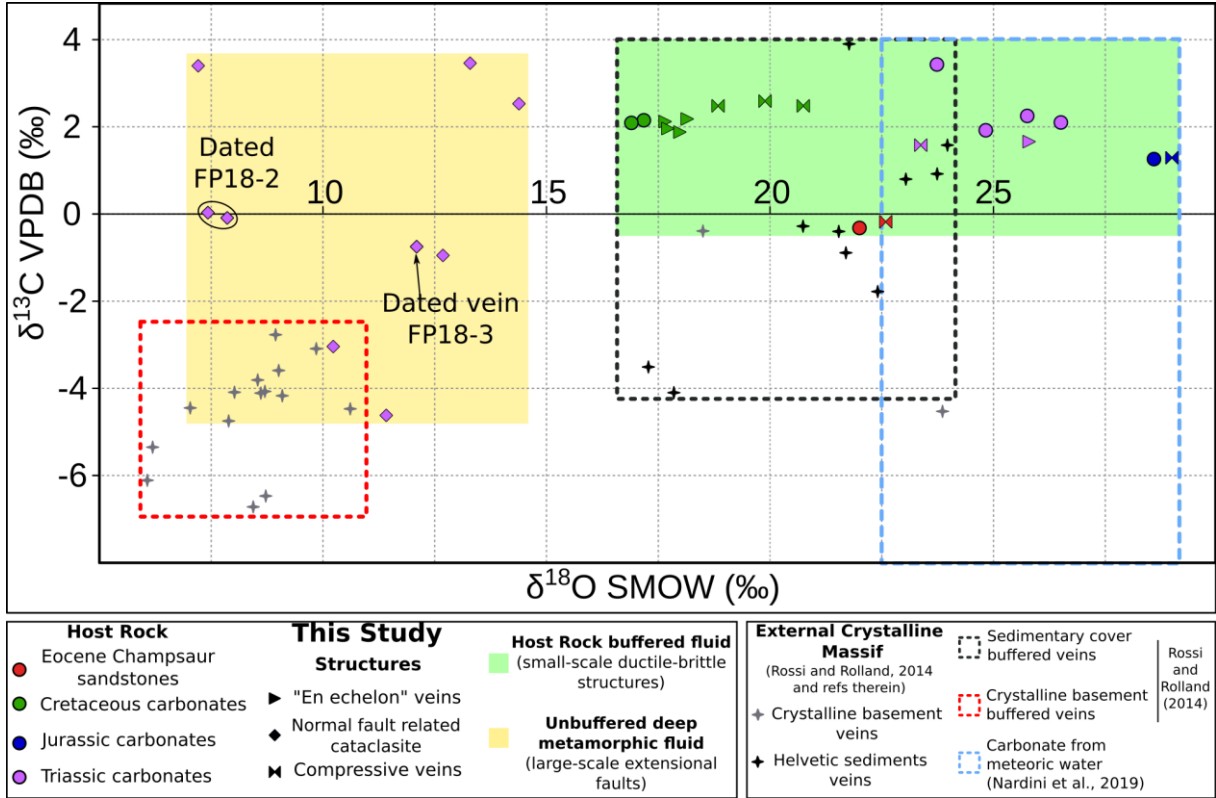

***Fig. 6.*** *Stable isotopic data from samples indicated on Fig. 2. Domains represented by dashed red, black and blue*
*lines are from the literature (Nardini et al., 2019; Rossi and Rolland, 2014 and references therein). The coloured*
*green domain corresponds to veins associated to brittle-ductile structures. These veins show similar isotopic*
*compositions as their host rocks. The orange domain features the signature of cataclased normal fault samples,*
*which show a different isotopic composition as compared to their host rock, and are similar to deep metamorphic*
*fluids (e.g., Crespo-Blanc et al., 1995; Rossi and Rolland, 2014; Rolland and Rossi, 2016).*

*4.3. Calcite LA-ICPMS U-Pb dating results*
Petrographic analysis has been complemented by screening using LA-ICP-MS on 24 thin-sections from
samples of 7 locations around the PFT related to shortening and extensional structures. Among these,
20 screened samples show high common lead contents, and sometimes higher lead to uranium intensity
signals. U-Pb dating of such carbonates with high lead concentrations remains highly challenging,
especially for very young samples. However, four samples (samples FP18-2, 3 and FP19-12A&B
described in section 4.1 and supplementary data) from the Tournoux normal fault site bear sufficient
$^{238}U$ (~0-8.5 ppm for FP18-3A&B and ~0-4.5 ppm for FP18-2B and FP19-12A&B), and $^{206}Pb$, $^{207}Pb$
(~0-1.9 ppm for FP18-3A&B and ~0-13.1 ppm for FP18-2B and FP19-12A&B).

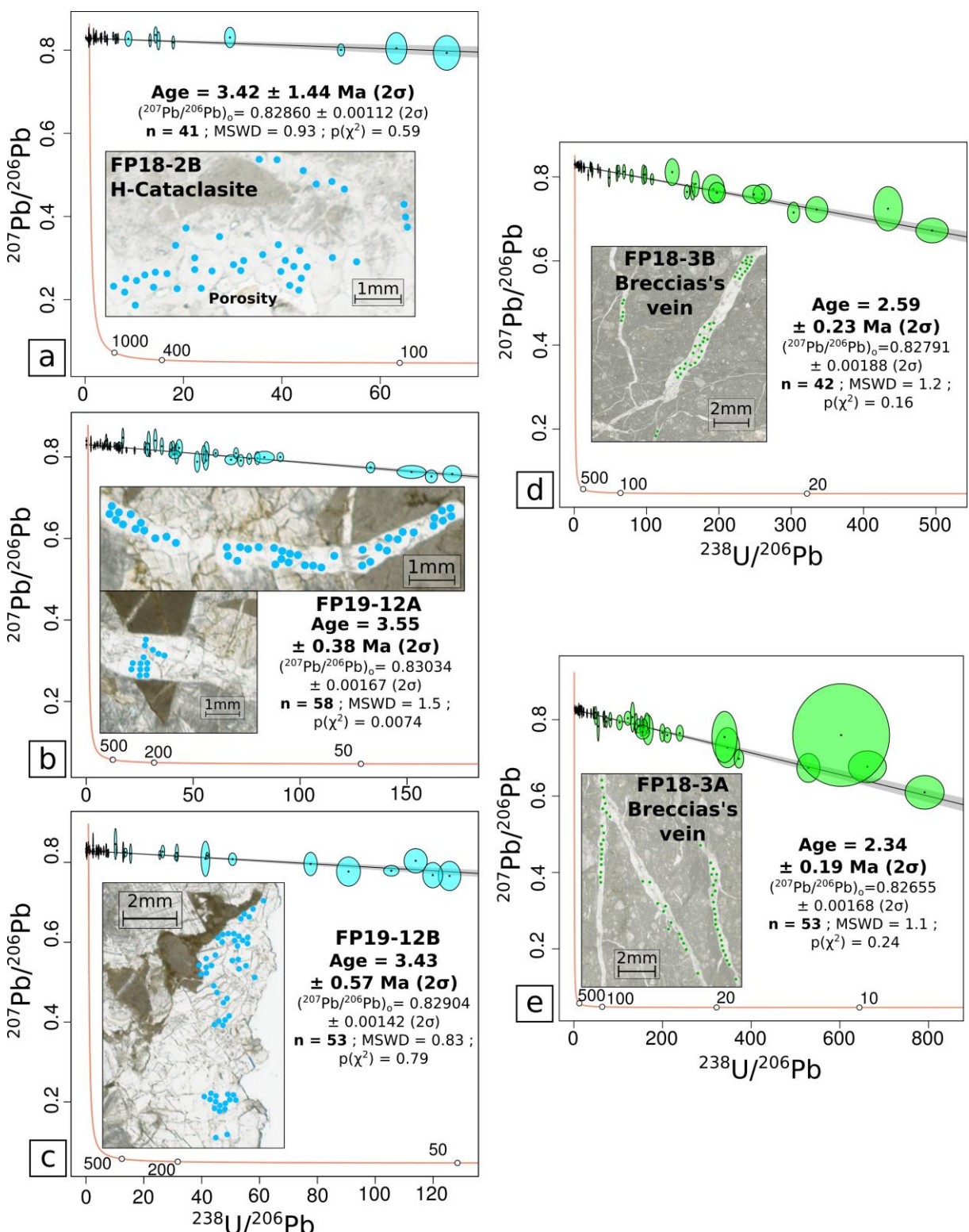

**Fig. 7.** *Tera-Wasserburg concordia plot of (a) Highly cataclased sample FP18-2 calcite filling (b) and (c) sample*
*FP19-12A veins and FP19-12A 'clean calcite' filling (d) and (e) sample FP18-3 veins, and corresponding maps*
*of sampled spots (150 μm). MSWD: Mean Square Weighted Deviation. An additional error propagation tied to*
*WC-1 standard uncertainty is taken into account.*

Lead contents are based on NIST614 intensities and uranium contents are based on WC-1 intensities
(Jochum et al., 2011; Roberts et al., 2017; Woodhead et al., 2001), giving measurable and significant
radiogenic signal. Five ages have been obtained on these four samples (Fig. 7).
A first group of ages of ~3.5 Ma is represented by three samples. The cataclasite 'clean calcite' infill
(sample FP18-2B; Fig. 5) gives age of 3.42±1.44 Ma (n=41, MSWD=0.93). This quite large uncertainty
is due to a relatively moderate U/Pb variability and the resulting low radiogenic signal measurable in
this sample. Samples FP19-12A&B give two similar within-error ages for the vein calcite and 'clean
calcite' infill, of 3.55±0.38 (n=58, MSWD=1.5) and 3.43±0.57 (n=53, MSWD=0.83), respectively.
A second group of ages of ~2.5 Ma is obtained on different cross-cutting veins of the latest generation
of sample FP-18-3 (Fig. 5), represented by slightly younger, but distinct out of error margins, ages of
2.59±0.23 Ma (n=42, MSWD=1.2; FP-18-3B in Fig. 7b) and 2.34±0.19 Ma (n=53, MSWD=1.1; FP-18-
3B in Fig. 7c). The higher spread in U/Pb ratios measured in these two latter ages results in more precise
and robust ages. These two age groups obtained on extensional faults connected to the PFT highlight
for the first time at least two phases of deformation constrained out of error bars: a first phase of brittle
deformation forming the cataclasite at 3.5±0.4 and one or two discrete brittle events at, or comprised
within, 2.6±0.2 and 2.3±0.2 Ma. These ages show that the sated conjugated faults have beed active for
at least 1 Myr, and are featured by only several datable events, representing co-seismic motions on the
faults.

**5. Discussion**
Onset of extensional tectonics in the Alps has remained a topic of debate for the last 20 years. A Miocene
age has been proposed for the onset of the extensional activation of the PFT based on AFT datings on
both sides of this major fault, i.e. in the Pelvoux External Crystalline Massif and in the Champsaur
sandstones to the west and in the Briançonnais zone to the east (Tricart et al., 2001; 2007; Beucher et
al., 2012). The Briançonnais zone corresponds to the east hanging wall compartment of the PFT. In this
compartment, AFT ages ranging from 30 Ma to 20 Ma are interpreted as the exhumation age of this area
related to the compressional activity of the PFT during the Alpine collision, which motion is constrained
by direct $^{40}Ar/^{39}Ar$ dating on phengite at 35-25 Ma (Simon-Labric et al., 2009; Bellanger et al., 2015).
To the west (footwall of the PFT), the AFT ages range from 13 Ma to 4 Ma in the Pelvoux External
Crystalline Massif (Beucher et al., 2012), and from 9 to 4 Ma in the Champsaur sandstones (Tricart et
al., 2007), and are interpreted as the extensional reactivation of the PFT by these latter authors. As the
AFT dates record an exhumation age associated with cooling below ~100°C (Ault et al., 2019), they
may not correspond to an age of PFT activity but rather record an erosion process that is related to both
climatic and tectonic processes (e.g., Champagnac et al., 2007). Sternai et al. (2019) suggest that vertical
movement in the Western Alps may be mainly ascribed to erosion and deglaciation (Nocquet et al.,
2016) and may also include a significant mantle convection component (Salimbeni et al., 2018).
However, the External Crystalline Massifs exhumation was also driven by frontal thrusting, activated
during middle Miocene at the western front of these massifs (Boutoux et al., 2015) and by strong
erosional processes that enhanced exhumation since the Late Miocene (Cederbom et al., 2004). Along
the PFT, younger AFT and phengite $^{40}Ar/^{39}Ar$ ages of ~10 Ma were obtained on the Plan de Phasy
(Guillestre) metagranite mylonites (Tricart et al., 2007; Lanari et al., 2014). These ages have been
interpreted as the result of hydrothermal fluid circulation, which may be linked to tectonic activity of
the High-Durance Fault System. However these fluid circulations may be passive through the PFT
network and may not correspond to extension onset. Therefore, the age of PFT activity remains
unconstrained and requires some direct datings. In the following discussion, we show how absolute U-
Pb dating of fracture infill calcite brings quantitative time constraints on PFT fault movement.

*5.1. Deformation and scale of fluid flow in the brittle-ductile structures*
The measured $\delta^{18}O$ and $\delta^{13}C$ isotope ratios of veins from brittle-ductile structures are close or similar to
their host rocks, and remain close to the field of carbonates precipitated from meteoric water (section
4.2). Based on several studies in the frontal parts of Alpine orogens (Smeraglia et al., 2020; Nardini et
al., 2019), these isotope signatures are thought to be representative of meteoric water inflow from the
most superficial domains. Three important parameters are involved to control this surface-derived fluid
regime: (i) lack of large-scale structures (ii) pressure-solution microstructures (evidence of local fluid)
(iii) presence of a shallow impermeable clay-rich layers which isolate upper crust from more deeply-
rooted systems (section 4.1. and Fig. 3). Rossi and Rolland (2014) report similar stable isotope
signatures in the Mont Blanc External Crystalline Massif sedimentary cover (Helvetic schists). There,
the vein calcites bear similar stable isotope values as the host Helvetic schists, which is in agreement
with the fluids to have equilibrated with their host rocks in a closed system with low fluid/rock ratios
(Rolland and Rossi, 2016). In our study, observations of veins show that they were closely related to
schistosity acting as a stylolithic dissolution surface (section 4.1). This observation is consistent with
local fluid interactions and equilibrium with the host rock, resulting from a pressure-dissolution-
recrystallization transfer mode (e.g. Passchier and Throw, 2005). Based on this, we suggest that the
external fluid signature was buffered by the host rock signature. These fluid compositions show that 'en-
echelon' veins are linked to an early deformation, where the porosity was still not connected by the fault
network (Fig. 3). In such a system, the veins kept the host rock signature and no crustal-scale fluid flow
circulation is evidenced.



*5.2. Scale of fluid flow in the brittle extensional structures*

Major (> metre-scale width) faults are related to shallower, or higher stress contexts (e.g. Passchier and Throw, 2005). The isotopic composition of calcite that crystallised in these brittle extensional faults is significantly different from their host rock (section 4.2; Fig. 6). Indeed, calcites related to these major faults have $\delta^{18}O$ lower than 10 ‰ from their host rock and a $\delta^{13}C$ ranging between -5 to 4 ‰ PDB (while the $\delta^{13}C$ ratio of Trias host rock is of 2 ‰). This signature is similar to that of exogenous metamorphic fluid origin (Crespo-Blanc et al., 1995; Rossi and Rolland, 2014). The observed CL pattern of calcites also argues for variations in the fluid composition, between the different veins and progressively within a given vein. Similar signatures are recorded in the Mont Blanc External Crystalline Massif shear zones and veins in a similar structural context (Rossi et al., 2014). There, a similar spread of $\delta^{13}C$-$\delta^{18}O$ values is observed in the marginal part of the crystalline basement, at the contact with the Helvetic schists. This spread is interpreted as a mixing between fluids flowing down through the sedimentary cover and upwards fluids originating from shear zones in the Mont Blanc Massif's central (Rolland and Rossi, 2016). The chemical signature of calcite veins in the Massif Central shear zones is correlated to a Mg-K-rich metasomatism, both arguing for $CO_2$-bearing fluids representative of a deep source, which is rooted in the mantle via vertical shear zones (Rossi et al., 2005). This deeply rooted fluid cell is also suggested by fluids significantly hotter (150-250 °C) than their host-rock at ca. 10 Ma along vertical faults in Belledonne Massif, which are in continuity with the central Mont Blanc Massif shear zones (Janots et al., 2019). Indeed, deep metamorphic fluid circulation is in good agreement with a crustal-scale fluid pathway which is activated during the extensional motion of the PFT, connected to the Rhône-Simplon right-lateral fault (Bergemann et al., 2019; 2020). This crustal-scale network suggests that extensional faults are in-depth connected to the PFT, when it was reactivated as a detachment. Deep connection with the PFT crustal scale structure (e.g. Sue et al., 2003) would allow fluid circulation from interface of European slab with the deep subduction/collisional metamorphosed prism. In our study, the isotopic dataset shows a significant difference between the deep fluids signature recorded by the Mont Blanc veins (Rossi and Rolland, 2014; Rolland and Rossi, 2016) and the compositions of the veins related to brittle-ductile structures (Fig. 6). This variability suggests a mixing process between the local fluids trapped in the early extensional (closed system) and these exogenous fluids from a deep crustal origin.

*5.3. Timing of PFT extensional inversion*

All ages obtained from the investigated Tournoux normal fault scarps, give direct time constraints on the final stages of extensional slip, and are interpreted as a minimum age for the extensional reactivation of the PFT. The oldest event is the formation of the highly deformed cataclasite calcite filling/veins ~3.5 Ma (3.4±1.5 Ma, 3.6±0.4 and 3.4±0.6). This calcitic cementation occurred directly after the main

cataclastic deformation event and before the late cross-cutting veins. Latter cross-cutting veins gave the
same ~2.5 Ma age, with two within-error dates of 2.6±0.3 and 2.3±0.3 Ma. The ~3.5 Ma and ~2.5 Ma
do not represent the same slip event on the fault. It is noteworthy that all these ages are calculated
assuming secular equilibrium in the U-series decay chain. As fluids are generally characterized by an
excess in $^{234}$U with respect to $^{238}$U, resulting in an excess of radiogenic $^{206}$Pb, the calculated ages should
be considered as maximum ages (see for example Walker et al., 2006). The magnitude of the offset ages
due to initial $^{234}$U/$^{238}$U disequilibrium can be significant and the true age could be younger by several
hundreds of thousands of years. In the present case, it was not possible to carry out classical isotopic
analyses of uranium by isotopic dilution to measure any detectable residual $^{234}$U/$^{238}$U disequilibrium
because of the size of the carbonate phases. It could be hazardous to speculate on the initial $^{234}$U/$^{238}$U
disequilibria of the fluids, but the quite high uranium concentrations (up to the ppm level) observed in
analysed minerals of samples FP-18-2 & 3 (Fig. 7) are likely indicative of an oxidizing environment and
thus of a moderate initial $^{234}$U excess (Walker et al., 2006). To assess the impact of this excess on the
final age, we have tested various initial $^{234}$U/$^{238}$U activity ratios ranging between 1 to 2 as illustrated in
Figure 8. For an initial ($^{234}$U/$^{238}$U) activity ratio of 2, the true age is lower by about ~370 ka. The obtained
ages assuming an initial ($^{234}$U/$^{238}$U) ratio of 1 are thus regarded as maximum ages.
As they remain undeformed, the latter veins are considered as the youngest tectonic slip along the fault.
Furthermore, the geometry of the Tournoux normal fault reguarding the PFT position indicates that this
normal fault was connected to the PFT, which acted as a detachment Zone (Fig. 9). Thus, it may
represent the paleo-HDFS seismogenic zone, which was later exhumed in the footwall part of the active
extensional fault. Main activity of this paleo-fault can be bracketed between 3.4-2.2 Ma based on the
above results.

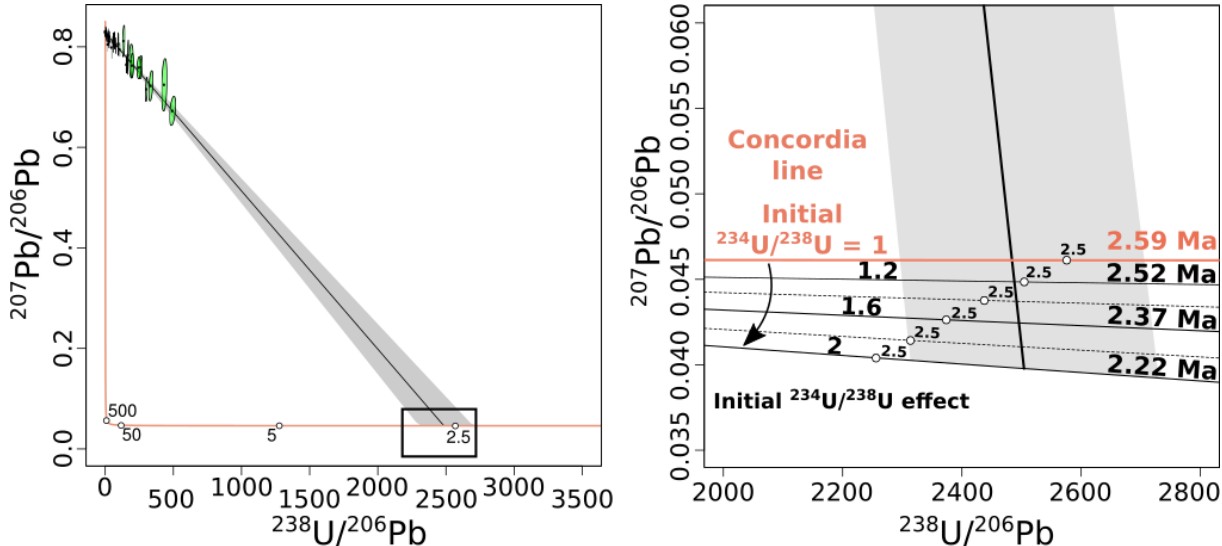

**Fig. 8.** *Impact of the initial $^{234}$U excess on the final age estimation. Several initial $^{234}$U/$^{238}$U activity ratios have been tested ranging between 1 to 2. This spread in initial $^{234}$U/$^{238}$U leads to an age difference of 0.37 Ma. The obtained U/Pb age of 2.59 Ma, assuming equality of $^{234}$U and $^{238}$U contents is thus a maximum age.*

*5.4. Evolution of PFT through time*

The structural and dating results presented in this paper, combined with the literature on PFT footwall and hanging wall exhumation lead to the following reconstitution of its evolution (Fig. 9).

The investigated PFT paleoseismic zone is located 3 to 10 km west of the active HDFS seismogenic zone. Nowadays, the extensional deformation is mainly localised on one active fault and mostly occurs mostly at 3 to 8 km depth (Sue et al., 2007; Mathey et al., 2020). This study gives insights into the uplift rate and lateral displacement of the High-Durance Fault System footwall and hanging wall since the passage of the investigated paleo-PFT through the upper boundary of the seismogenic crust some 2-3.5 Ma ago. Since then, the PFT hanging wall, represented by the active extensional deformation front of the HDFS was significantly shifted eastward, while its footwall was uplifted up to 3 km (Fig. 9). This leads to a mean vertical tectonic motion on the order of > 1 mm.yr$^{-1}$ for the footwall compartment of PFT on this period of time. This rate is consistent with the vertical GPS rates measured for the Pelvoux External Crystalline Massif (Nocquet et al., 2016; Sternai et al., 2019).

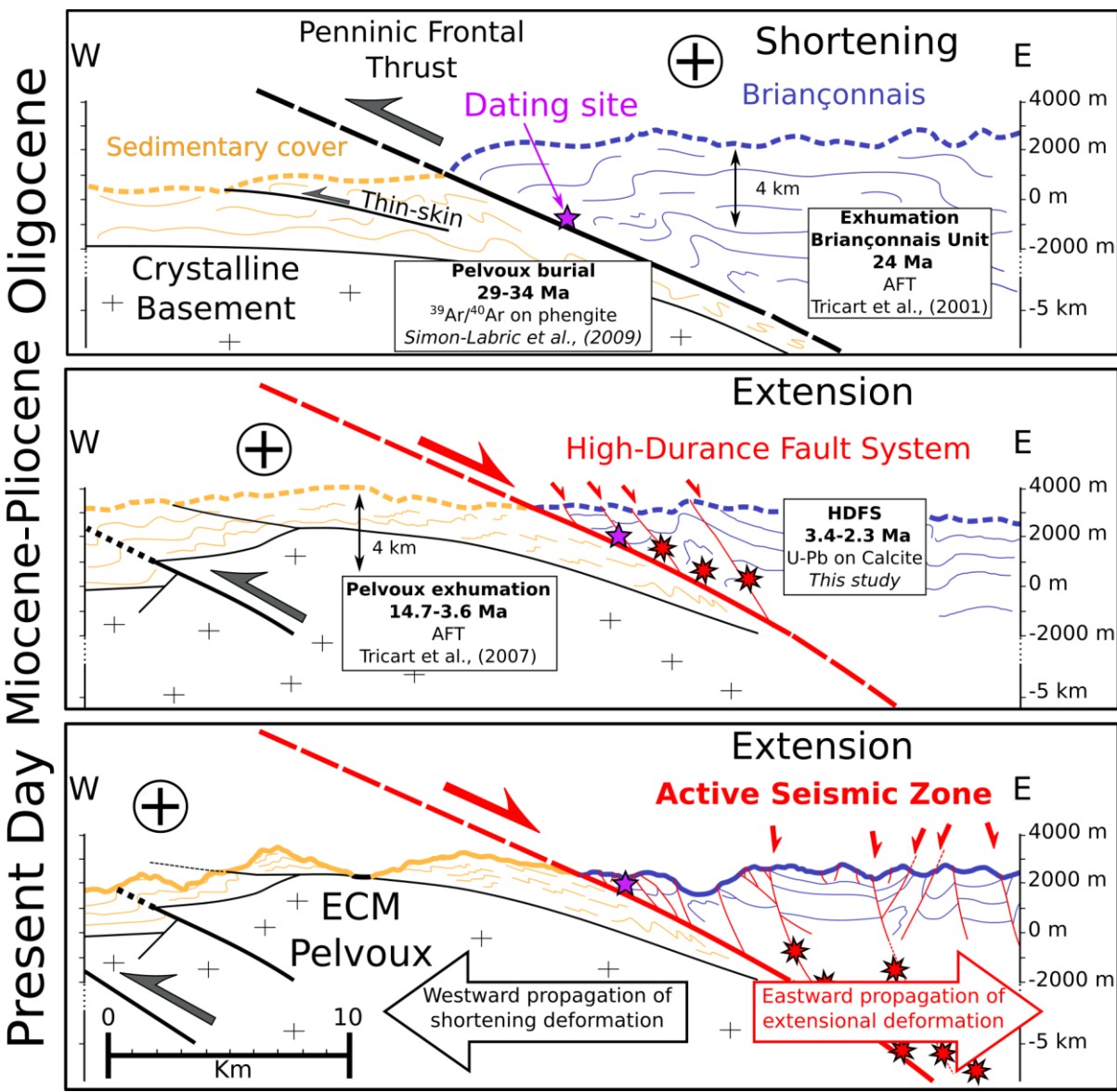

**Fig. 9.** *Evolutionary geological cross-section sketch of PFT (modified from Tricart et al., 2006).* **a,** *Compressional activation of the PFT resulting in joint External Crystalline Massifs burial and Briançonnais exhumation during the Oligocene.* **b,** *Extensional reactivation of the PFT and setting up of the High-Durance Fault System during the Pliocene as evidenced in this study. At this point the dated extensional fault passes through the upper boundary of the seismic zone at ca. 2-3 Ma.* **c,** *At present-day, compressional deformation has migrated westward (frontal part of External Crystalline Massifs, since c. 15 Ma) and extensional seismic activity of the High-Durance Fault System is recorded at shallow depth 3-10 km east of the studied paleoseismic zone.*

Our data support the hypothesis that the present HDFS is the result of eastward shifting of extensional deformation, accommodated by successive jumps on several faults. Faults were likely active on a scale of ≥1 Myr before becoming inactive. Calcite U-Pb ages obtained on the Tournoux scarp constrain co-seismic motion on two conjugate faults. The two age groups obtained on these extensional faults connected to the PFT highlight at least two phases of deformation, at 3.5±0.4 Ma and one or two discrete

brittle events at, or comprised within, 2.6±0.2 and 2.3±0.2 Ma, which gives insights into the long-term activity of the of at least 1 Myr, but with only several datable events, which argues for an apparent contradiction. Indeed, co-seismic displacement on the fault suggest a significant magnitude for the related earthquake (Wells and Coppersmith, 1994), which is apparently incompatible with the very few datable motions. This gives some weight to a deformation regime which may alternate long phases of creeping on the fault plane, without any brittle deformation, with very rare phases of brittle deformation. The vertical uplift and exhumation of the Pelvoux External Massif since 3.5 Ma may thus mainly result from the cumulated fault motion on these several fault segments. These data are thus in agreement with a significant tectonic component in the measured uplift signal of External Crystalline Massifs, which is in agreement with a clear difference in uplift rates measured between ECMs and Internal Alps (Nocquet et al., 2016; Sternai et al., 2019).

We support the hypothesis that the HDFS is the result of the eastward extensional deformation shift and the successive activation of faults, which incrementally participated at the exhumation of the western (Pelvoux) footwall side of the fault system.

## 6. Conclusion

Significant constraints on the evolution of fault systems can be acquired by coupling stable isotopic analysis and U-Pb dating on calcite. These methods have been successfully applied to unravel the tectonic reactivation of the PFT for the first time. Five U-Pb ages on calcite have been obtained on extensional fault structures connected to the PFT, gave two distinct groups of ages of 3.5±0.5 Ma for the main deformation phase represented by the cataclasite calcite cement, cross-cut by later discrete phases represented by mm-large veins dated from 2.6±0.3 to 2.3±0.3 Ma. The 3.5 Ma age represents a minimum age for the onset of extensional brittle reactivation of the PFT. Earliest extensional ductile-brittle structures cannot be dated due to low uranium contents and low U/Pb ratios. Associated to those two (ductile and brittle) deformation stages, stable isotopic ratios of carbon ($\delta^{13}$C) and oxygen ($\delta^{18}$O) of calcite samples collected within the kilometre-scale extensional faults show an evolution from a closed to an open fluid system. The isotopic signature of fluids related to the brittle deformation stage corresponds an open system due to the activation of a crustal-scale fluid circulation cell when the HDFS developed in connection with the deeper PFT deeper structure. The fluids associated to this open system show a deep crustal/mantle signature similar to that measured along the PFT across the Alpine arc. This deeply rooted upward fluid circulation occurred when extensional fault activity was connected to the PFT reactivated as a detachment, which suggests a crustal-scale extensional reactivation at this stage. These constraints on PFT fluid regime are the first direct evidence for a transition towards a crustal-scale fluid regime at the onset of brittle extensional reactivation in the Alps. The direct ages of PFT motion give insights into the long-term incremental displacement of the HDFS footwall, and Pelvoux

Massif exhumation, which corresponds to its passage through the upper part of the seismogenic zone, at
a mean rate of $> 1$ mm.yr$^{-1}$ in the last 3 Ma.

**Acknowledgements**
This work forms part of first author's Ph.D. funded by the BRGM in the frame of the RGF project. The
CEREGE group is supported by 2 French "Investissements d'Avenir" fundings: the EQUIPEX-ASTER-
CEREGE and the Initiative d'Excellence of Aix-Marseille University - A*Midex, through the DatCarb
project. We wish to thank Fayçal Soufi for his help in sample preparation.
Many thanks are due to Alfons Berger, anonymous reviewer and Giancarlo Molli for their constructive
comments which improved the manuscript.

**Author contributions**
AB, YR and SS wrote the manuscript and all authors discussed the results and contributed to the final
article. TD supported AB for map creation and cross-sections. YR, SS, TD, CG and AB participated to
field trip sampling. AB did the sample petrographic characterization with optical microscope and
cathodoluminescence. NG, AG and PD led U-Pb dating with AB. BB and AN supervised AB for stable
isotopes analysis, for results interpretation and protocol application respectively.

**Supplementary Materials**
Suppl. Mat. Table S1. Sample locations and descriptions.
Suppl. Mat. Fig. S1. Tournoux's scarp general view.
Suppl. Mat. Fig. S2. Field photographies FP19-12 site.
Suppl. Mat. Fig. S3. La-ICPMS elemental maps, FP18-2B.
Suppl. Mat. Fig. S4. FP19-12B thin section with map localisation.
Suppl. Mat. Fig. S5. La-ICPMS elemental map, FP19-12B.

Suppl. Mat. Table S2. U-Pb on calcite La-ICPMS data.

## Table 1: Isotopic composition of analysed calcites

|  | N°Sample | $^{13}\delta C$ PDB | $^{18}\delta O$ SMOW |
|---|---|---|---|
| *Host Rock* | FP18-1A | 2,15 | 17,18 |
|  | FP18-1A | 2,09 | 16,90 |
|  | FP18-4 | 1,92 | 24,83 |
|  | FP18-7 | 2,25 | 25,76 |
|  | FP18-9 | -0,32 | 22,00 |
|  | FP18-10 | 1,26 | 28,59 |
|  | FP18-11 | 2,1 | 26,51 |
|  | FP18-13 | 3,43 | 23,73 |
| *Early veins (V1)* | FP18-1B | 2,59 | 19,89 |
|  | FP18-1C | 2,48 | 20,74 |
|  | FP18-1C | 2,48 | 18,84 |
|  | FP18-9 | -0,18 | 22,59 |
|  | FP18-10 | 1,29 | 28,99 |
|  | FP18-11 | 1,58 | 23,37 |
| *En-echelon veins (V2)* | FP18-1A | 2,12 | 17,65 |
|  | FP18-1A | 2,18 | 18,15 |
|  | FP18-1B | 1,96 | 17,71 |
|  | FP18-1D | 1,88 | 17,98 |
|  | FP18-5 | 1,66 | 25,80 |
| Cataclasite infill *(V2)* | FP18-2A | 0,03 | 7,43 |
|  | FP18-2B | -0,09 | 7,86 |
|  | FP18-3B | -4,62 | 11,41 |
|  | FP18-3B | -0,75 | 12,10 |
|  | FP18-6 | -3,04 | 10,23 |
|  | FP18-6 | -0,95 | 12,68 |
|  | FP18-13 | 3,46 | 13,29 |
|  | FP18-13 | 2,53 | 14,38 |
|  | FP18-13 | 3,4 | 7,21 |

**Table 1.** *Stable isotope data from host rocks, calcite veins and cataclasite fillings of extensional faults.*

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
