# Peer review of "Extensional reactivation of the Penninic Frontal Thrust 3 Ma ago as evidenced by U-Pb dating on calcite in fault zone cataclasite."

_Solid Earth, 2020_

## Referee Comment (RC1) · Alfons Berger (Referee) · 28 Sep 2020

**Review**:  "*Extensional reactivation of the Penninic Frontal Thrust 3 Ma ago as evidenced by U-Pb dating on calcite in fault zone cataclasite*" **by Bilau et al.**

**General comments:**

The paper gives constrain on the age of vein formation in (or near) the Penninic Frontal Thrust (=PFT). In addition, well performed stable isotopes of vein carbonates are used for reconstructing the fluid source of such veins. The data are all well performed and documented. The main problem may be a nomenclature problem. For many geologist, the name "PFT" is reserved for the thrusting of Pennine units on top of the foreland. This is not the topic of the paper. As you state in your abstract the vein formation and the extension is somehow related "High-Durance Fault System" (see also your Line 97). In contrast, the introduction gives more an overview on the PFT, but not on the High-Durance Fault System. In other words, the introduction should give higher relevance to the Pliocene/Pleistocene extensional tectonics (e.g., Sue et al. 2007) instead on the Oligocene thrusting. The spatial overlap of the PFT and the High-Durance Fault should be described in detail at the beginning.

**Detail comments:**

Line 19:     add "so called" or introduce somehow the "High Durance extensional fault"
Line 29/30: This sentence may be to complex for most readers. "*Extension is caused by compression, which is propagating*..." ???
Line 81:     You may add "*Agard et al. (2002)*"
Line 83:     better see "Rubatto and Hermann (2001)"
Line 90:     The sentence is misleading. In Simon Labric et al. 2009 there is also white mica from the PFT itself.
Line 96:     see also constrains for the deformation history of the Briançonnais and Subbriançonnais in Ceriani and Schmid (2004) and related literature (Ceriani, Bucher etc)
Line 113:   please add a reference (or a figure)
Line 299:   FT ages only record cooling, which require somehow also erosion at the end. It is difficult to constrain the tectonics out of the FT data, specially if the ages are overlapping ages of both sides of the PFT.

This is nice little piece of work
Best Regards
Alfons Berger

---

## Referee Comment (RC2) · Anonymous Referee #2 · 3 Nov 2020

REVIEW of the paper SE-202-119

Extensional reactivation of the Penninic Frontal Thrust 3 Ma ago as evidenced by U-Pb dating on calcite in fault zone cataclasite by A. Bilau et al.

This paper presents the first attempt to date the extensional reactivation of the so-called Penninic Frontal Thrust PFT in the Southwestern alpine arc. The paper is concise, well organized and well written. It brings new important quantitative data for the understanding of the late-alpine tectonics in the Southwestern Alps as a whole. A weakness of the paper concerns the relevant bibliography, that must be improved, mainly in the introduction and discussion parts.

[Figure]

Indeed, I do recommend the acceptation of the ms. in Solid Earth, after minor revision.

I detail hereafter step-by-step recommendations.

abstract - line 29-31: the discussion on the coeval extension in the internal zones and compression propagation in the external zone is not well constrained/dated and is not properly address in the discussion part of the ms. a specific paragraph could be added in the discussion. However, it is not a key point of the paper, and could be discarded.

1. introduction - line 46: does the PFT really acted as a "plate boundary"? eventually discuss and/or present the structural relations between the Briançonnais and the external zone. - line 48: also refer to Sue and Tricart (1999, Eclogae Geol. helv. ; 2003, Tectonics) for the reactivation of the PFT in extension and the description of the regional fault system. - line 51 also refer to Sternai et al. (2019, ESR) for the isostatic/buoyancy forces discussion.

2. Geological setting - line 64 67: the concept of "plate boundary" implies to consider the briançonnais zone as a single (micro)plate. I do think that this point deserves a longer analyze, specifically in terms of paoleogeography. Quote also Tricart, (1984, Am. J. Sci) for the PFT top-to-the-west thrusting history. - line 68: Zhao et al (2016) is an important reference in the frame of this ms. but not on the nappe-related structure. Write a specific sentence for the lithospheric structure seen by Zhao et al. - line 80-82: also quote Agard (2002 J. Metam. Geol). - line 94-95: also quote the synthesis of Bertrand and Sue (2017, Swiss J. Geosci.) - line 97-101: the overall seismotectonic local framework in the study area, including geodesy, should be better exposed. See for instance the recent paper by Mathey et al., (2020, GJI). the same matter arises in the discussion part. - line 96: Note that the very first reports of the brianconnais' seismicity has been published by Rothè (1941). The seismotectonic regional frame is first described by Sue et al. (1999, JGR); these references could be added. - line 101: the Jenatton et al (2007) and Leclère et al. (2012)'s works focused on the Ubaye swarm, to the South of the study area, which actually occurred West of the PFT, in

relation with fluid circulation. This thematic could be discussed in the ms., but in a specific paragraph, as these works are not directly connected to the PFT reactivation. - line 120: the same Oreac section has been described by Sue and Tricart (1999, Eclogae Geol. Helv.) in term of brittle deformation and related paleostress.

3. Sampling strategy and analytical method - this part is well organized, precise and informative.

4. Results fig4a: could you provide the corresponding photography? give also a close-up location map of the samples (smaller scale than fig.2). - line 243 and following: better explain the stable isotope results, for a non-specialist. - line 262-263: the comparison with the Mont-Blanc ECM is very interesting. It must be better developed in the discussion part. In the present form, the last sentence of the paragraph is unuseful. Either discard it, or (better) develop a bit more. - line 275-276: better explain this sentence (re-write). - line 277-283: these ages are very good regarding the questions still under debate on the overall late extension thematic. Moreover, they represent the core of the paper. I would advise to better underline the quality and novelty of these pretty young ages. - Fig7 could be enlarged. The figures and words embedded in the panels are not legible.

5. Discussion the overall discussion is written with a pretty affirmative tone. I suggest the authors to use more careful words in their interpretations. - line 319320: precise and rewrite the 3 points (i) (ii) and (iii) in a more logical way. - line 332-333: this sentence is unclear. rewrite and develop a bit the concept you wanna describe. - line 340-345: the comparison with the Mont-Blanc ECM deserves to be better developed. I would suggest to writte a complete paragraph on this comparison, eventually supported by a new specific figure, including a map view of the related MB vs. Brianconnais contexts. Concerning the MB's exhumation processes, quote at least Seward and Mancktelow (1994, Geology). - Line 347, together with Zhao et al (2016), the references to the ECORS profile and related interpretations regarding the PFT at depth must be quoted (e.g. Mugnier et al., BSGF 1993). I also suggest to quote the ECORS

cross-section re-assessed by Schmid and Kissling (2000, Tectonics). - line 380: the fault dated in the ms. "may" represent a paleo-HD fault. It is still an interpretation. - line 389-400: this very small paragraph on "evolution through time" (indeed from c.a. 3 Ma up to now and the active deformation) must be better developed and improved. A map of the active deformation at the local scale could be interesting. The paragraph should integrate discussion on the uplift, which is not restricted to the ECM, but also affect the inner area (Nocquet et al. 2016; Sternai et al., 2019), together with the extension seen both in geodesy (e.g. Walpersdorf et al., 2015, J. Geodyn) and looking at the focal mechanisms of earthquakes (Sue et al. 1999 JGR ; 2007 IJES). Indeed, such a discussion should bring the gap between the current activity of the Briançonnais area, which is well constrained, and the "late alpine" faulting, which is now well dated by the present paper.

---

## Author Response (AR2)

Yann Rolland
Associate Professor at Université Savoie Mont Blanc. Laboratoire EDYTEM - UMR5204 Bâtiment « Pôle Montagne », 5 bd de la mer
Caspienne, F-73376 Le Bourget du Lac cedex, France.
Tel : 0033-4 83 61 85 86, Yann.Rolland@univ-smb.fr
Antonin Bilau
PhD candidate at Edytem at Université Savoie Mont Blanc antonin.bilau@univ-smb.fr

[Figure]

**Object:**  20th November 2020

Article #  se202-119 resubmission
to *Solid Earth*

To: *Solid Earth* Editorial Office
C. Sue editor

Dear editor(s),
Dear Christian Sue,

We are pleased to re-submit our manuscript entitled:
**"Extensional reactivation of the Penninic Frontal Thrust 3 Ma ago as evidenced by U-Pb dating on calcite in fault zone cataclasite",**
 authored by Antonin Bilau, Yann Rolland, Stéphane Schwartz, Nicolas Godeau, Abel Guihou, Pierre Deschamps, Benjamin Brigaud, Aurélie Noret, Thierry Dumont and Cécile Gautheron.

We are glad of the very positive and constructive reviews, and wish to thank the careful reading of the two reviewers. We followed most reviewer corrections, where possible, and in case of disagreement we explained why we kept our former interpretation. We also incorporated two new ages that comfort the previously obtained age of 3.5 Ma, as we obtained them after the submission process and they nicely confirm this age with a much narrower error bar.

We hope that you will find that the corrections have significantly improved the manuscript in a way to make it suitable for your journal.

Thanks also for the efficient editorial handling.

With kind regards, and on behalf of co-authors,

Antonin Bilau and Yann Rolland

**Response to reviewer 1 (Alfons Berger):**

**General comments:**

The paper gives constrain on the age of vein formation in(or near) the Penninic Frontal Thrust(=PFT). In addition,well performed stable isotopes of vein carbonates are used for reconstructing the fluid source of such veins. The data are all well performed and documented. The main problem may be a nomenclature problem. For many geologist, the name "PFT" is reserved for the thrusting of Pennine units on top of the foreland. This is not the topic of the paper. As you state in your abstract the vein formation and the extension is some how related "High-Durance Fault System"(see also your Line 97). In contrast, the introduction gives more an overview on the PFT, but not on the High-Durance Fault System. In other words, the introduction should give higher relevance to the Pliocene/Pleistocene extensional tectonics (e.g., Sue et al. 2007) instead on the Oligocene thrusting. The spatial overlap of the PFT and the High-Durance Fault should be described in detail at the beginning.

Thanks to Alfons Berger for his consideration and positive review. We reworked some sentences in order to precise the duality link between High-Durance Fault System and the PFT. However, here, we follow the common understanding of PFT in the western Alps, at the boundary with the Pelvoux Massif. Most authors argue that the HDFS is the expression of the PFT reactivation as a normal structure in Briançonnais zone (following especially, Sue and Tricart, 1999, 2003). So, we don't think this might lead to some confusion between the two. To ensure a good comprehension of the meaning of the two systems PFT/ HDFS, we made efforts to clarify this view as much as possible.

**Detail comments :**

Line 19:add "so called" or introduce somehow the "High Durance extensional fault"
OK. Modified.

Line 29/30:This sentence may be to complex for most readers. "Extension is caused by compression, which is propagating..."???
OK, we modified the text for more clarity.
"This reactivation may result from the westward propagation of the compressional deformation toward the External Alps, combined to the exhumation of External Crystalline Massifs. In this context, the exhumation of the dated normal faults is linked to the eastward translation of the HDFS seismogenic zone in agreement with the present day seismic activity."

Line 81:You may add "Agard et al. (2002)"
Read & Added.

Line 83:better see "Rubatto and Hermann (2001)"
Read & Added.

Line 90:The sentence is misleading. In Simon Labric et al. 2009 there is also whitemica from the PFT itself.
OK, corrected. We agree.

Line 96:see also constrains for the deformation history of the Briançonnais and Subbriançonnais in Ceriani and Schmid (2004) and related literature (Ceriani,Bucher etc).
Read & Added.

Line 113:please add a reference(or a figure).
Added, Fig.2.

Line 299: FT ages only record cooling, which require some how also erosion at the end. It is difficult to constrain the tectonics out of the FT data, specially if the ages are overlapping ages of both sides of the PFT.
We agree with you, FT ages are not direct datings of tectonic motions, and their signal can be misleading in this matter. However, as these were the only data that existed before to constrain PFT extensional motion, and as ages obtaines on both sides of the PFT do not overlap there is a suggestion of PFT activity that is worth mentioning.
* * *
**Response to reviewer 2:**

Thanks a lot to Rev. 2 for his careful reading and advise about our paper. We followed his propositions in detail.

**abstract**
line 29-31: the discussion on the coeval extension in the internal zones and compression propagation in the external zone is not well constrained/dated and is not properly address in the discussion part of the ms. a specific paragraph could be added in the discussion. However, it is not a key point of the paper, and could be discarded.
OK. Reworked.

**1. introduction**
line 46: does the PFT really acted as a "plate boundary"? eventually discuss and/or present the structural relations between the Briançonnais and the external zone.
Right, modified : "as the major tectonic structure".

line 48: also refer to Sue and Tricart (1999, Eclogae Geol. helv. ;2003, Tectonics) for the reactivation of the PFT in extension and the description of the regional fault system.
Read & Added.

line 51 also refer to Sternai et al. (2019, ESR) for the isostatic/buoyancy forces discussion.
Read & Added.

**2. Geological setting.**
line 64-67: the concept of "plate boundary" implies to consider the briançonnais zone as a single (micro)plate. I do think that this point deserves a longer analyze, specifically in terms of paoleogeography. Quote also Tricart, (1984,Am. J. Sci) for the PFT top-to-the-west thrusting history.
Right, modified : «as the major tectonic structure".

line 68: Zhao et al (2016) is an important reference in the frame of this ms. but not on the nappe-related structure. Write a specific sentence for the lithospheric structure seen by Zhao et al.
Completed with « Schmid and Kissling 2000, Lardeaux et al., 2006, Malusà et al., 2017". And Ceriani et al. for the nappe structure.

line 80-82: also quote Agard (2002 J. Metam. Geol).
Read & Added.

line 94-95: also quote the synthesis of Bertrand and Sue (2017, Swiss J. Geosci.)
Read & Added.

line 97-101: the overall seismotectonic local framework in the study area, including geodesy, should be better exposed. See for instance the recent paper by Mathey et al., (2020, GJI). the same matter arises in the discussion part.
Read & Added.

line 96: Note that the very first reports of the brianconnais'seismicity has been published by Rothè (1941). The seismotectonic regional frame is first described by Sue et al. (1999, JGR); these references could be added.
OK, very well. These refs have been added.

line101: the Jenatton et al (2007) and Leclère et al. (2012)'s works focused on the Ubaye swarm, to the South of the study area, which actually occurred West of the PFT, with fluid circulation. This thematic could be discussed in the ms., but in a specific paragraph, as these works are not directly connected to the PFT reactivation.
Right, removed.

line 120: the same Oreac section has been described by Sue and Tricart (1999,Eclogae Geol. Helv.) in term of brittle deformation and related paleostress.
Read & Added.

**3. Sampling strategy and analytical method**
this part is well organized, precise and informative.
Fine. Thank you.

**4. Results**
fig4a: could you provide the corresponding photography? give also a close-up location map of the samples (smaller scale than fig.2).
Modified, the original photography is in Fig.3c.

line 243 and following:better explain the stable isotope results, for a non-specialist.
Addition of formulation of equation (1): $\delta^{13}C$ calculation. And "The ratio of carbon and oxygen isotopes is related to the parental fluid of calcite and can be used as a fluid tracer."

line 262-263: the com-parison with the Mont-Blanc ECM is very interesting. It must be better developed in the discussion part. In the present form, the last sentence of the paragraph is unuseful. Either discard it, or (better) develop a bit more.
OK, discussion and links with the Mt-Blc have been developed.

line 275-276: better explain this sen-tence (re-write).
Reworked and completed. The details pertaining to analytical proc. have been better explained in the corresponding section.

line 277-283: these ages are very good regarding the questions still under debate on the overall late extension thematic. Moreover, they represent the core of the paper. I would advise to better underline the quality and novelty of these pretty young ages.

Thanks for this comment. We complemented this section and reworked the conclusion to highlight those ages and corresponding fluid history better.

Fig7 could be enlarged. The figures and words embedded in the panels are not legible.

OK, this has been done. In addition, 2 more ages coming from new sample in the same area have been added, and elemental map see supplementary data.

**5. Discussion**

the overall discussion is written with a pretty affirmative tone. I suggest the authors to use more careful words in their interpretations.

Taken into account, sentences have been rewritten in a less affirmative way.

line 319-320: precise and rewrite the 3 points (i) (ii) and (iii) in a more logical way.

Reworked and completed: « (i) lack of large-scale structures (ii) pressure-solution microstructures (evidence of local fluid) (iii) presence of a shallow impermeable clay layer which isolate surface and deep systems".

line 332-333: this sentence is unclear. rewrite and develop a bit the concept you wanna describe.

OK, rewritten.

line 340-345: the comparison with the Mont-Blanc ECM deserves to be better developed. I would suggest to write a complete paragraph on this comparison, eventually supported by a new specific figure, including a map view of the related MB vs. Brian-connais contexts. Concerning the MB's exhumation processes, quote at least Sewardand Mancktelow (1994, Geology).

This comparison has been precised, with some more details on the MB context. However, besides this is clear that fluids have a similar signature, the age of structures is different (15 Ma in Mt Blc) and so is the context (extensional here, compressional Mt Blc), so we don't think the comparison has to be so much extended.

Line 347, together with Zhao et al (2016), the ref-erences to the ECORS profile and related interpretations regarding the PFT at depth must be quoted (e.g. Mugnier et al., BSGF 1993). I also suggest to quote the

ECORS cross-section re-assessed by Schmid and Kissling (2000, Tectonics).

OK, these refs have been added.

line 380: the fault dated in the ms. "may" represent a paleo-HD fault. It is still an interpretation.

Added.

line389-400: this very small paragraph on "evolution through time" (indeed from c.a. 3 Ma up to now and the active deformation) must be better developed and improved. A map of the active deformation at the local scale could be interesting. The paragraph should integrate discussion on the uplift, which is not restricted to the ECM, but also affect the inner area (Nocquet et al. 2016; Sternai et al., 2019), together with the extension seen both in geodesy (e.g. Walpersdorf et al., 2015, J. Geodyn) and looking at the focal mechanisms of earthquakes (Sue et al. 1999 JGR ; 2007 IJES). Indeed, such a discussion should bring the gap between the current activity of the Briançonnais area,which is well constrained, and the "late alpine" faulting, which is now well dated by the present paper.

OK, we agree, we have enhanced this part.
* * *
**Response to Topical Editor:**

Both reviewers agreed since the first run of revision that the manuscript is an interesting contribution suitable to be pubblished in the SE Special issue. I'm completely in agreement with the reviewers especially now that the manuscript is also improved taking into account the suggestions and comments of both reviewers.
I recommend therefore acceptance of this very nice paper maybe considering some changes included in the annotated pdf (text and one figure).

I suggest to change the "stress-related" term of "compression" with "contraction or shortening" if as in your case is used in combination with extension (indeed a "strain -related" term).
Note however that the Figure 9 stage "present day" compression is written with only 1 s. compresion please change

Thanks, the manuscript has been corrected accordingly to the consideration for the use of the term "shortening" instead of "compression".

[revised manuscript text omitted]